# The Role of Cardiopulmonary Exercise Testing in Hypertrophic Cardiomyopathy

**DOI:** 10.3390/medicina59071296

**Published:** 2023-07-13

**Authors:** Lidija Mikic, Arsen Ristic, Natasa Markovic Nikolic, Milorad Tesic, Djordje G. Jakovljevic, Ross Arena, Thomas G. Allison, Dejana Popovic

**Affiliations:** 1Division of Cardiology, Clinical and Hospital Center Zvezdara, 11120 Belgrade, Serbia; 2Division of Cardiology, University Clinical Center of Serbia, 11000 Belgrade, Serbia; 3Faculty of Medicine, University of Belgrade, 11000 Belgrade, Serbia; 4Institute for Health and Wellbeing (CSELS), Faculty of Health and Life Sciences, Coventry University, Coventry CV1 2DS, UK; djordje.jakovljevic@newcastle.ac.uk; 5Translational and Clinical Research Institute, Faculty of Medical Sciences, Newcastle University, Newcastle NE1 7RU, UK; 6Department of Physical Therapy, College of Applied Health Sciences, University of Illinois at Chicago, Chicago, IL 60607, USA; 7Department of Cardiovascular Medicine, Mayo Clinic, Rochester, MN 55905, USA; 8Department of Pediatric and Adolescent Medicine, Division of Pediatric Cardiology, Mayo Clinic, Rochester, MN 55905, USA

**Keywords:** cardiopulmonary exercise testing, hypertrophic cardiomyopathy

## Abstract

This review emphasizes the importance of cardiopulmonary exercise testing (CPET) in patients diagnosed with hypertrophic cardiomyopathy (HCM). In contrast to standard exercise testing and stress echoes, which are limited due to the ECG changes and wall motion abnormalities that characterize this condition, CPET allows for the assessment of the complex pathophysiology and severity of the disease, its mechanisms of functional limitation, and its risk stratification. It is useful tool to evaluate the risk for sudden cardiac death and select patients for cardiac resynchronization therapy (CRT), cardiac transplantation, or mechanical circulatory support, especially when symptomatology and functional status are uncertain. It may help in differentiating HCM from other forms of cardiac hypertrophy, such as athletes’ heart. Finally, it is used to guide and monitor therapy as well as for exercise prescription. It may be considered every 2 years in clinically stable patients or every year in patients with worsening symptoms. Although performed only in specialized centers, CPET combined with echocardiography (i.e., CPET imaging) and invasive CPET are more informative and provide a better assessment of cardiac functional status, left ventricular outflow tract obstruction, and diastolic dysfunction during exercise in patients with HCM.

## 1. Hypertrophic Cardiomyopathy—Pathophysiology to Target

Hypertrophic cardiomyopathy (HCM) is a disease of the cardiac muscle defined by the presence of increased left ventricular (LV) wall thickness that cannot be explained exclusively by abnormal loading conditions [1]. The latest guidelines [1,2] suggest that HCM is a disease defined by its phenotype, which, in adults, is defined as a wall thickness ≥15 mm in one or more LV myocardial segments. However, more limited hypertrophy (13–14 mm) can be diagnostic when present in family members of a patient with HCM or in conjunction with a positive genetic test. Hypertrophic cardiomyopathy is the most common inherited heart disease with a prevalence of 1:200 to 1:500 in the general population, equally distributed by sex [1,2]. In comparison to newer AHA/ACC guidelines, the ESC recommends a broader approach to the term HCM; many genetic and acquired disorders have a similar phenotype to LV hypertrophy. In adolescents and adults, 40–60% of cases are caused by autosomal dominant sarcomeric gene mutations, 25–30% have unknown etiology, and 5–10% of cases hide disorders such as amyloidosis, Anderson Fabry, Danon cardiomyopathy in adults, mitochondrial and metabolic diseases, RASopathies, sarcoid, hemochromatosis, and drug toxicity [1]. The AHA/ACC have different approaches to defining the condition and consider HCM specific to a historically well-described genetic disease caused by the mutation of one of the sarcomeric proteins dominantly affecting the septum [2]. Additional diagnostic challenges arise from secondary LVH, such as hypertensive cardiomyopathy, athletes’ heart, hypertrophy due to valvular or subvalvular LV obstructive lesions, or obstruction after antero-apical infarction and stress cardiomyopathy [2].

Most patients with HCM are asymptomatic for an extended period and can have a normal life expectancy [1,2,3]. As the disease progresses, patients can experience exertional fatigue and dyspnea, syncope, angina, and palpitations. The pathophysiological mechanisms behind these symptoms are complex and consist of LV outflow tract impairment, diastolic dysfunction, chronotropic incompetence, and microvascular and even peripheral muscle changes [4]. Almost thirty years ago, Lele and other authors described the inability to increase cardiac stroke volume as the most important cause of exertional incompetence in patients with HCM [5]. A combination of septal hypertrophy and the systolic anterior motion (SAM) of the mitral leaflet causes a systolic LV outflow pressure gradient and resting obstruction in approximately 30% of HCM patients [4,6,7]. That number rises to almost half of the patients during exercise—out of 11,672 patients studied from 69 different research articles, an LVOT gradient >30 mmHg was present at baseline in 31.4% of cases and increased to 49% during exercise [7]. Shah and colleagues found that 54 out of 87 symptomatic patients with no previously documented LVOT had latent LV outflow tract obstruction [8]. Impaired relaxation and reduced LV compliance increase LV telediastolic pressure, which worsens the often-coexisting mitral regurgitation [4]. There is an increase in left atrial pressure and eventually left atrial volume, resulting in atrial arrhythmias and retrograde secondary pulmonary hypertension [1,2,4]. Chronotropic incompetence may also play an important role in reduced exercise capacity in HCM patients [9,10], potentially resulting from a remodeling of the sinoatrial node or impaired cell signaling [10]. Microvascular dysfunction in a hypertrophied myocardium and subendocardial ischemia may cause chest pain [11]. Myofibrillar disarray, fibrosis, and a remodeling of the LV are risk factors for life threatening ventricular arrhythmias [12,13]. Some patients eventually develop heart failure (HF) with a preserved LV ejection fraction (HFpEF) phenotype due to LV diastolic dysfunction, or HF with a reduced LV ejection fraction (HFrEF) phenotype as a consequence of LV wall thinning and LV dilatation [3].

## 2. Exercise Testing to Assess HCM

Cardiopulmonary exercise testing (CPET) combines standard exercise testing measurements (i.e., blood pressure, ECG, and symptom assessment) with ventilatory expired gas analysis. The use of CPET provides enhanced information on the severity of the disease and its mechanism(s) of functional limitation compared to that of standard exercise testing and stress echocardiography, which are limited due to the ECG changes and wall motion abnormalities that are common in HCM in the absence of coronary artery disease [2].

A cycle ergometer or a treadmill is an acceptable exercise modality for CPET in patients diagnosed with HCM [2]. CPET is used to quantify cardiorespiratory fitness, discover the pathophysiological mechanism underlying exercise intolerance and formulate a function-based prognostic stratification [14,15]. CPET provides a detailed and comprehensive way to approach the complex pathophysiology of HCM and can be a useful tool in assessing prognosis and treatment, especially in recognizing patients with a higher risk for sudden cardiac death and HF development [7,16]. CPET is also a useful tool in differentiating HCM from other forms of LV hypertrophy, such as athletes’ heart, as well as in the evaluation of athletes with a confirmed diagnosis of HCM [17]. Moreover, CPET is used to monitor therapeutic efficacy in this patient population [18].

A number of monitored and calculated CPET parameters may be helpful in targeting HCM diagnosis and assessing its risks, including but not limited to the following: (1) blood pressure; (2) HR and ECG changes (3) maximal or peak oxygen consumption (VO_2_); (4) percentage of age- and sex-predicted maximal/peak VO_2_; (5) ventilatory anaerobic threshold (VAT); (6) oxygen O_2_ pulse (i.e., amount of oxygen extracted by tissues per heartbeat); (7) ventilatory efficiency (i.e., minute ventilation (VE)/carbon-dioxide (CO_2_) production slope); (8) partial pressure of end-tidal CO_2_ pressure (P_ET_CO_2_); and (9) pattern of breathing, or respiratory reserve at the end of exercise (BR) [15,16].

## 3. Aerobic Capacity—Peak VO_2_

Peak VO_2_ is a central CPET parameter and has great prognostic value in HCM [14,15,19]. Percentages that are <80% of the predicted VO_2_ for the patient’s age, gender, and height are considered abnormal [20], which is common in the HCM population. Cui et al. analyzed 752 patients diagnosed with HCM and found that the mean peak VO_2_ was 18.0 mLO_2_·kg^−1^·min^−1^, which was 60% of the predicted value. The primary causes attributed to low peak VO_2_ were impaired cardiac output (73.7%), limited HR reserve (52.0%), and obesity (48.2%) [19]. Interestingly, a resting LV outflow tract gradient correlated poorly with peak VO_2_, which was confirmed by recent studies as well [7,21]. Patients were followed for a median of 9.0 years and it was found that an adjusted peak VO_2_ and an abnormal O_2_ pulse increase were independently associated with long-term survival after myectomy. Through analyzing more than 50 publications in HCM cohorts and, collectively, 11,672 HCM patients (48 ± 14 years old, of which 65.9% were men and 34.1% women), Bayonas-Ruiz et al. found that the mean peak VO_2_ was 22.3 ± 3.8 mLO_2_·kg^−1^·min^−1^ and concluded that it is a disease with an exercise capacity reduced by at least 20% that of what is expected in an apparently healthy individual [7]. There was no significant difference in peak VO_2_ between patients with severe vs. milder hypertrophy, nor between subgroups with obstructive vs. non-obstructive HCM. However, the mean peak VO_2_ was 6.2 mLO_2_·kg^−1^·min^−1^ less in patients who died during the follow up period compared to those who survived [7]. Masri et al. studied 1005 HCM patients (50 ± 14 years, 64% male) and found that a peak VO_2_ < 50% of the predicted value was independently associated with overall mortality and SCD [22]. Similarly, in a study of 156 patients with HCM (mean follow-up at 27 ± 11 months), Finocchiaro et al. concluded that a peak VO_2_ <20 mLO_2_·kg^−1^·min^−1^ or <80% of the predicted value was associated with a worse prognosis [23]. Coats et al. followed 1898 patients with HCM (47 ± 15 years, 67% male) who underwent CPET at the beginning of the follow up period during the time period between 1998 and 2010. In this study, aerobic capacity was shown to be a strong predictor for death and heart transplantation (primary end points). The patients who had a peak oxygen consumption of ≤15.3 mL/kg/min had a 14% or 31% chance of dying or having a heart transplant in a 5 or 10 year period, respectively [16]. In a prognostic study by Sorajja et al., with a 4.0 ± 3.2 year follow up, a population of 182 minimally symptomatic patients with obstructive HCM (53 ± 15 years; 65% male, 35% female) who had a peak VO_2_ less than 60% of the predicted value had a 41% chance of dying or having severe symptoms in a 4 year period [24]. The fact that, in the earlier mentioned study by Masri et al., only 8% of patients with HCM achieved >100% age-gender predicted peak VO_2_ [22] can be a valuable clinical measure in the differentiation between physiologic LV hypertrophy in athletes and HCM, especially in the so called “gray zone” [25]. Recently, 58 athletes with HCM evaluated at the Mayo Clinic were found to have a mildly reduced exercise capacity (83% of predicted); however, a reduced peak VO_2_ did not correlate with symptom status or clinical outcomes [17]. The fact that two recent clinical trials that evaluated the usage of the new therapeutic drug mavacamten in the treatment of obstructive HCM, EXPLORER, and PIONEER HCM used the change in peak VO_2_ as both primary and secondary endpoints, or only the secondary endpoint, indicates the importance of functional testing in the management of HCM [26,27].

## 4. Ventilatory Anaerobic Threshold

The ventilatory anaerobic threshold represents the moment in exercise where ventilation starts to exponentially rise due to the transition toward an anaerobic metabolism with a relative increase in carbon dioxide (CO_2_) production compared to VO_2_ [14]. The ventilatory anaerobic threshold typically occurs between 45% to 65% of the measured peak VO_2_ in healthy untrained individuals and, in general, at a higher percentage in trained subjects. The later VAT occurs (i.e., the closer the VAT and peak VO_2_ approximate), the greater one’s capacity for submaximal exertion will be [14]. The already cited study of Coats et al. found that for each 1 mLO_2_·kg^−1^·min^−1^ reduction in VAT, there was a 29% increase in the risk of death or heart transplantation in an HCM cohort [16]. A recent meta-analysis by Bayonas-Ruiz et al. revealed an average VAT of 14 mLO_2_·kg^−1^·min^−1^ in patients with HCM and demonstrated significant prognostic value [27].

## 5. Oxygen Pulse

Oxygen pulse is viewed as a surrogate for stroke volume, and as such, a flattening of the O_2_ pulse curve during exercise is considered a strong prognostic parameter in all HF phenotypes [28]. In patients diagnosed with HCM, abnormal temporal behavior of the O_2_ pulse during exercise is strongly related to an inadequate stroke volume increase and correlates with a reduced peak VO_2_ and VAT, as well as an increased VE/VCO_2_ slope, identifying more advanced disease irrespective of the LV outflow tract obstruction [21]. In a study by Mapelli et al., 96 out of 312 patients with non-end-stage HCM (70% of which were males, aged 49 ± 18 years), that is, 31%, presented with abnormal O_2_-pulse temporal behavior, irrespective of their LVOTO values. 

## 6. Ventilatory Efficiency and P_ET_CO_2_

Ventilatory efficiency represents cardiopulmonary coupling through an assessment of VE and VCO_2_. A VE/VCO_2_ slope >30 is an established abnormal threshold in the general population [14]. In patients diagnosed with HCM, diastolic dysfunction can lead to a retrograde increase in pulmonary pressure [1,14]. In this situation, as VE increases, there is lack of a concomitant matching in pulmonary circulation (i.e., a lack of perfusion of normally unperfused areas due to increased pressures) and the VE/VCO_2_ slope is abnormally elevated; the degree of the VE/VCO_2_ slope reflects the degree of increased pulmonary pressure. The progression of pulmonary hypertension indicates a greater HCM severity. Similarly, another marker of ventilation–perfusion mismatch is P_ET_CO_2_, which is normally >37 mmHg in apparently healthy individuals at peak exercise. In the 2012 clinical recommendations for CPET in specific populations, Guazzi et al. made a semiquantitative classification of HCM severity using the following parameters: (1) the VE/VCO_2_ slope; (2) percentage of predicted peak VO_2_; and (3) P_ET_CO_2_. The greater the VE/VCO_2_ slope (or ratio during exercise) and the lower the percentage of predicted peak VO_2_ and P_ET_CO_2_, the greater the likelihood of increased pulmonary pressure and, with it, the severity of HCM [14]. In a study cohort of 623 HCM patients (49 ± 16 years old, 69% male, 3.7 year follow up), Magri et al. investigated CPET parameters that can predict SCD risk, finding a VE/VCO_2_ slope >31 to be the most accurate among CPET parameters in predicting the SCD end point (sensitivity 64%, specificity 72%) [29]. In an earlier mentioned study by Coats at el, the risk of death or heart transplantation was increased by 18% for each unit increase in the VE/VCO_2_ slope [16]. A VE/VCO_2_ slope >34 and left atrial enlargement were defined as the main predictors of overall mortality, heart transplantation, and deterioration to septal reduction in a study by Finocchiaro et al. [23]. In a study by Velicki et al. who analyzed a total of 41 patients with nonobstructive HCM who were recruited into the ongoing SILICOFCM study, the VE/VCO_2_ slope was found to be the most sensitive CPET parameter for gauging the therapeutic efficacy of a 16 week sacubitril/valsartan treatment [30].

## 7. Blood Pressure Response to Exercise

During both standard exercise testing and CPET, blood pressure measurement is essential and obligatory. Systolic blood pressure should rise, with an upper normal threshold of 210 mmHg in males and 190 mmHg in females during exercise, while diastolic blood pressure is normally unchanged or slightly decreased [14]. An abnormal blood pressure response to exercise (ABPRE) is defined as hypotension or the inability to increase blood pressure (>20 mm Hg) with exercise [31], which can be explained by impaired stroke volume and/or vagal driven vasodilatation [4]. This parameter is well recognized as a marker of hemodynamic instability and increased risk factor for SCD in patients with HCM younger than 40 years [1,2,7,31]. In a metanalysis by Bayonas-Ruiz et al., the mean prevalence of ABPRE in HCM patients was 20%, and this proportion was higher in severe groups (>20 mm) than in mild hypertrophy groups (17.9% vs. 13.6%) [7]. Both European and American guidelines point out ABPRE as a risk factor for SCD in young adults with HCM [1,2]. However, ABPRE is not included in the provided risk scores for SCD and indications for ICD implantation in primary prevention [1]. This is explained by the fact that the positive predictive accuracy of this parameter is too low to allow for the identification of a high-risk patient based only on an abnormal test result. On the other hand, it has a high negative predictive accuracy for HCM-related mortality in the absence of other risk factors and can be used as a screening test for the identification of low-risk patients [31]. Circulatory power (i.e., peakVO_2_ x peak systolic BP) was shown to be useful in monitoring therapeutic efficacy, as demonstrated in the EXPLORER-HCM mavacamten study [18].

## 8. Chronotropic Competence

Electrocardiogram monitoring, typically 12 leads, is an inevitable part of both standard exercise testing and CPET, providing the ability to monitor HR and rhythm by an appropriately qualified health professional, which is of particular importance in patients diagnosed with HCM [14]. The minimal normal HR response to exercise, for an individual not on beta-blockade, is approximately 80–85% of maximal predicted HR for a given age (usually calculated as 220—age) [14]. HR increases linearly with time, work rate, and VO_2_ at approximately 10 beats per 3.5 mLO_2_·kg^−1^·min^−1^ [14,15]. HR recovery should be ≥12 beats within the first minute of exercise cessation [15]. Chronotropic incompetence plays a role in the exercise limitation of patients with HCM, likely due to remodeling of the sinoatrial node or impaired cell signaling [9,10]. Efthimiadis et al. studied 68 HCM patients (age 44.8 ± 14.6 years, 45 males), who underwent CPET and discovered that half of the cohort demonstrated chronotropic incompetence [9]. This abnormal response was associated with a higher functional disability, history of atrial fibrillation, signs of fibrosis at nuclear magnetic resonance imaging, and the usage of antiarrhythmic drugs, especially beta-blockers [9]. During a 4.2 year follow up of 681 patients with HCM (48 ± 16 years old, 68% male), Magri et al. found that a peak HR < 70% was a risk factor for HF related events and found this abnormal response held prognostic significance in HCM patients [32].

## 9. Arrhythmias

At the beginning of 20th century, the AHA Guidelines cited CPET as a relative contraindication in patients with HCM due to arrhythmias that could possibly occur during testing [14]. In the following years, it was proven that CPET is safe to perform in the HCM population and that the risk of developing fatal arrhythmia during exercise is relatively low (~0.2%) [13,33]. However, if altered rhythm, ectopic foci, or ST changes are verified by ECG monitoring during exercise, SCD risk is elevated [14]. The occurrence of arrhythmia is higher in young adults with HCM and the presence of ventricular arrhythmias is significantly associated with adverse events, as shown in the data from the metanalysis by Bayonas-Ruiz et al. [7]. Non-sustained ventricular arrhythmia and unexplained syncope are listed as individual risk factors for SCD in HCM and are used with other parameters in SCD risk calculation and the assessment of ICD implantation indications [1]. Gimeno et al. studied 1380 patients with HCM (42 ± 15 years; 62% male; mean follow-up at 54 ± 49 months). Twenty-seven patients (40 ± 14 years) had NSVT [24] or ventricular fibrillation (VF) [3] during exercise. These patients had more severe hypertrophy and larger left atria. The occurrence of NSVT/VF was more common in males (22 (81.5%) were male) [13]. An HCM-impacted myocardium, which progresses from hypertrophy to fibrosis and eventually dilation with microvascular alterations, increases the susceptibility to ventricular ectopic activity [34]. The fact that HCM-related fatal ventricular arrhythmia is the single most common cause of SCD in young competitive athletes (35%) further emphasizes the clinical value of CPET in all athletes and identifying asymptomatic individuals at risk [35].

## 10. Respiratory Dyssynchrony and Exercise Oscillatory Ventilation

Respiratory dyssynchrony (RD) is defined as a reduced ventilatory efficiency (i.e., an elevated VE/VCO_2_ slope) or exercise oscillatory ventilation (EOV) [36]. Exercise oscillatory ventilation is a pattern of breathing described as cyclic fluctuation of VE and expired gas during exercise that persists for at least 60% of the exercise test with an amplitude ≥ 15% of the average resting value [16,37]. It is still not well understood and can vary in its amplitude and frequency of oscillation, as well as duration. It is a marker of poor prognosis in patients with HF [16]. It is believed that it is caused by dysfunction in the alveolar–capillary unit, as well as peripheral and central chemo/baroreceptors [38]. In 2021, Potratz et al. studied 132 patients with hypertrophic non-obstructive cardiomyopathy (HNCM) (51 ± 17 years old; male 71%) who underwent CPET and proposed a new hypothesis that respiratory dyssynchrony can have a prognostic impact on patients with HNCM, recognizing EOV as a potential new factor in risk stratification. They found a significant association between a composite endpoint (i.e., death, heart transplantation, or implantation of a ventricular assist device) and the presence of EOV [36].

## 11. Left Ventricular Outflow Tract Obstruction and CPET

Hypertrophic cardiomyopathy was previously described as hypertrophic obstructive cardiomyopathy due to its common phenotype of septal hypertrophy and LV outflow tract obstruction [6]. LV outflow tract obstruction is defined as a systolic peak in the Doppler pressure gradient ≥30 mm Hg due to mitral–septal interaction [1,4]. It is present in approximately 30% of patients with HCM at rest. That number goes up to 50% during exercise [7]. As previously mentioned, Bayonas-Ruiz et al. analyzed 69 publications including 11,672 patients with HCM, and they showed that LVOT was present at rest in 31.4% of cases and increased to 49% during exercise [7]. An LV outflow tract gradient of ≥50 mmHg is hemodynamically important, usually causes symptoms, and, thus, represents a threshold in treatment decision making regarding invasive septal reduction procedures [39]. Sorraja et al. recognized the severity of thee LV outflow tract gradient at rest as an individual prognostic factor for death and severe adverse events after a 4.0 ± 3.2 year follow up of 182 patients with minimally symptomatic HOCM [24]. In this study, LVOT obstruction at rest was present in 96 patients (53%) and latent in 86 patients [24]. Interestingly, in a study by Mapelli et al., patients were grouped by degree of LVOT (72% of patients with LVOTO <30; 10% between 30 and 49 and 18% ≥50 mm Hg), and the results showed that none of the three analyzed CPET parameters (% of predicted peak VO_2,_ O_2_-pulse, and VE/VCO_2_ slope) were correlated with the degree of LVOTO.

In the European Society of Cardiology (ESC) Guidelines, CPET (or a standard exercise test when unavailable) has a class IIb recommendation and should be considered in symptomatic patients undergoing LV outflow tract invasive reduction procedures [1,21,40]. Recent AHA/ACC guidelines [2] suggest CPET imaging or exercise stress echocardiography for the detection and quantification of dynamic LV outflow tract obstruction in symptomatic and asymptomatic patients with HCM who do not have resting or a provocable outflow tract gradient ≥50 mmHg. Latent obstruction may be revealed by exercise stress echocardiography, which enlightens causes for exertional or postural syncope [41,42]. However, the assessment of the obstruction degree is not a simple act, being that meal-related hemodynamic changes may favor LV obstruction [43], and some medications, such as beta blockers, may lower the gradients during exercise [44].

## 12. CPET in Athletes with HCM

Sudden cardiac death in competitive athletes is a devastating and often preventable event. The most common cause of SCD in athletes is fatal arrhythmias in individuals with asymptomatic HCM [35]. Athletes tend to have physiologically increased LV wall thickness that can mask the true cause of hypertrophy and leave young individuals with HCM unrecognized during standard assessment [25]. The degree of physiologic hypertrophy is usually less than it is in HCM; however, in some cases, identifying diseased myocardium using only echocardiography and electrocardiography can be difficult. In “gray zone” cases, CPET is recommended [1,2,14,25]. Elite athletes with LV hypertrophy have significantly enhanced CPET responses, including higher peak VO_2_ and VO_2_ at VAT compared to athletes with HCM; only 8% of patients with HCM achieve >100% of the age–gender predicted peak VO_2_ [7,25]. A peak VO_2_ of >50 mLO_2_·kg^−1^·min^−1^ or more than 20% higher than the age–gender predicted peak VO_2_ is proposed as a standard for differentiating athlete’s heart from HCM [25]. ESC Preventive Cardiology guidelines suggest that a peak VO_2_ less than 84% of the predicted maximum, as well as the presence of symptoms, non-sustained ventricular tachycardia, ST-segment depression/T wave inversion during exercise, or ABPRE are indicative of pathological LV hypertrophy [45].

## 13. Recommendations

The latest European and American Guidelines include CPET as an important tool in the assessment and management of patients with HCM [1,2]. CPET is recommended in patients with nonobstructive HCM and advanced HF (NYHA functional class III to class IV) to quantify their degree of functional limitation [2]; moreover, it has class Ib recommendation and is a standard part of evaluation for patients with severe symptoms that are being considered for cardiac transplantation [46] or mechanical support [1,2]. CPET, if available, should be considered for the evaluation of the severity and mechanisms of exercise intolerance and changes in systolic blood pressure (class IIa) during the initial assessment of patients with HCM [1,2]. Additionally, CPET was suggested as one of the tests that should be considered in symptomatic patients undergoing invasive septal reduction procedures to determine the severity of exercise limitation (class IIa) [1]. CPET should also be considered in patients with uncertain symptoms and functional statuses, especially those planned for CRT [2]. Provokable obstruction that can be detected during CPET influences clinical decision making [2,47]. CPET results may also aid in clinical decision making in asymptomatic patients with comorbidities such as hypertension (e.g., hydration, diuretics, or vasodilators) [2]. In patients with uncertain clinical statuses, the assessment of functional capacity by CPET may be used to gauge therapeutic efficacy and guide decisions on therapy escalation or de-escalation [2]. The prognostic implications of CPET in HCM, as part of the initial evaluation, are qualified as a class II recommendation, although the evidence shows that there is an association among impaired CPET parameters, progression to advanced HF, and all-cause mortality in almost 3000 patients [2,4,47]. It may be considered every 2–3 years in clinically stable patients or every year in patients with worsening symptoms (class IIb) [1,2].

## 14. Recommendations for Exercise Prescription in Hypertrophic Cardiomyopathy

Previously, avoidance of competitive sports in HCM patients was strongly recommended. However, there are some new perspectives that are suggesting that, in some cases, participation in competitive sports can be safely managed. More data is needed to provide objective safety evaluations and decision making, for example, after septal reduction procedures or ICD implantation have been performed [17]. The ESC Preventive Cardiology guidelines suggest individual decision making and recommend (Class IIc) exercise testing to assess for the appearance of exercise-induced ventricular tachycardia, a significant increase in the LV outflow gradient (>50 mmHg), or ABPRE. These findings on an exercise test, in conjunction with symptoms (i.e., syncope), high ESC risk scores, and history of SCD, represent absolute contraindications for competitive sports in athletes with HCM [45].

Whereas guidelines do not strictly point out the use of CPET for exercise prescription, evidence shows that mild and moderate exercise programs in asymptomatic and symptomatic individuals with HCM are safe and lead to an increased functional capacity and improved quality of life [48]. Less data are available on the effects of vigorous exercise [48]. This topic was discussed in the most recent article by Bryde and other authors, who emphasized the importance of physical activity in HCM management [49]. In a study by Larsen et al., where almost half of the patients with HCM were overweight (45% of 510 patients had a BMI ≥ 30 kg/m^2^), the authors suggested that obesity can be a big attributing factor to a reduced exercise capacity unrelated to the disease itself [50]. In this condition, moderate physical activity at the same time helps weight loss, but also functional capacity improvement [50]. Individuals with HCM who regularly exercise have lower total cardiovascular mortality compared with those who do not [48].

## 15. Future of CPET in HCM

Although performed in fewer institutions, CPET combined with echocardiography (CPET imaging) is more informative in terms of cardiac functional status, LV outflow tract obstruction, and diastolic dysfunction during exercise in patients with HCM [15]. It also allows for a post-septal reduction follow up and for monitoring the response to the procedure [15]. There is a strong theoretical and practical rationale supporting the simultaneous gas exchange analysis with stress echocardiography [51,52,53]. They simultaneously provide a comprehensive understanding of overall functional status in the light of contractile function and relaxation of cardiac chambers, valve function, and left- and right-sided cardiac hemodynamics, enabling both strong diagnostic and prognostic information [54]. A huge advantage of this method is a noninvasive cardiac output calculation, which can be critical for adequate clinical insight in patients with HCM [54]. The appearance of B-lines in the lungs, detected by echocardiography during CPET, is a marker of pulmonary congestion during the effort, as shown in 2/3 of the total 128 patients diagnosed with HCM examined by Palinkas ED et al. [55]. In a study by Re F et al., exercise induced pulmonary hypertension was present in about one fifth of the total 182 patients diagnosed with HCM without evidence of elevated pulmonary pressures at rest and was associated with adverse clinical outcomes, suggesting that echocardiography combined with CPET may help physicians detect early stages of pulmonary hypertension, thus allowing for closer clinical monitoring and individualized therapies [56]. Another proven advantage of combined stress echocardiography and CPET is the ability to assess the right heart–pulmonary circulation uncoupling, which is likewise relevant in patients with pulmonary hypertension associated with HCM [57]. The estimation of the right ventricular contractile reserve by echocardiography may play an important role in the follow-up and therapeutic management of patients with HCM complicated with pulmonary hypertension [57]. Yet, large clinical trials have yet to be conducted.

The role of genetic testing over the CPET variables in HCM risk stratification was analyzed in a retrospective genotype–phenotype study of 371 patients screened for likely pathogenic/pathogenic (LP/P) genetic variants, at least for the main sarcomeric genes *MYBPC3* (myosin binding protein C), *MYH7* (β-myosin heavy chain), *TNNI3* (cardiac troponin I), and *TNNT2* (cardiac troponin T) [58]. The LP/P variant was associated with a more aggressive HCM phenotype; however, left atrial diameter, circulatory power (peak VO_2_*peak systolic blood pressure), and ventilatory efficiency were the only independent predictors of HF, whereas only left atrial size and circulatory power were predictors of sudden cardiac death after 5.1 years of follow up. This finding reaffirms the pivotal role of the CPET-echo-derived parameters in HCM risk stratification and indicates that the combination of genetic testing and CPET-echo may have adjunctive role in prognosis of patients with HCM.

Invasive CPET with the measurement of pulmonary pressure allows for a better assessment of exercise-induced diastolic dysfunction in HCM [20]. Simultaneous cardiac catheterization and CPET can provide essential hemodynamic data and identify patients who might benefit from septal reduction therapy and can also determine the presence of comorbidities, enable the monitoring of the response to medication and surgical interventions, estimate prognosis, and guide referrals for orthotopic heart transplantation [59]. Typical hemodynamic features predicting a favorable response to surgery include left atrial hypertension secondary to dynamic mitral regurgitation and acute development of an LV outflow tract obstruction [60]. The invasive assessment of hemodynamic adaptations to efforts combined with CPET, in patients with HCM, may provide greater insight into the primary mechanism of exertional symptoms, which might better predict responses to septal reduction therapy since it is a physiological rather than pharmacological provocation [61]. However, there are challenges to performing the accepted, more invasive, catheter-enabled measures in many clinical settings due to logistical complications, patient risks, and excessive cost. Equipping more laboratories around the world in this regard may contribute to the better diagnosis and management of HCM.

## 16. Conclusions

Collectively, CPET allows for comprehension of the complex pathophysiology of HCM and provides information on the severity of the disease, its mechanisms of functional limitation, and its risk stratification. It is useful tool to evaluate the risk of SCD and select patients for CRT, cardiac transplantation, or mechanical circulatory support, especially when their symptoms and functional status are uncertain. It may help in differentiating HCM from other forms of cardiac hypertrophy, such as athletes’ heart. Finally, it is used to guide and monitor therapy and for exercise prescription.

## Data Availability

Not applicable.

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
