# Peer review of "The Role of Cardiopulmonary Exercise Testing in Hypertrophic Cardiomyopathy"

_medicina, 2023, doi:10.3390/medicina59071296_

Round 1

Reviewer 1 Report

The publication is important from a practical perspective and provides a comprehensive overview of CPET role in hypertrophic cardiomyopathy.

Here are my comments:

1.     Add a diagram or a figure summarizing pathophysiological mechanisms underlying altered CPET parameters

2.     Line 201: besides the recent report from Wheeler et al., peakVO2 was also assessed in the first phase II RCT PIONEER HCM and in the first report of EXPLORER HCM. Please mention these 2 studies.

Minor editing of English language

Author Response

Reviewer 1

The publication is important from a practical perspective and provides a comprehensive overview of CPET role in hypertrophic cardiomyopathy.

Here are my comments:

  1. Add a diagram or a figure summarizing pathophysiological mechanisms underlying altered CPET parameters.

Thank you for the valuable recommendation. We added a sheme that summarizes most important pathophysiological mechanisms in HCM and their effects on CPET parameters.

Central Figure

  1. Line 201: besides the recent report from Wheeler et al., peakVO2 was also assessed in the first phase II RCT PIONEER HCM and in the first report of EXPLORER HCM. Please mention these 2 studies.

Thank you for this great suggestion, the fact that change in peakVO2 was used as a primary and secondary end point in these two important studies is a good indicator of the significance of CPET in HCM management. We included it in the section about aerobic capacity, page 3, as follows:

Line 145:

The fact that two recent clinical trials that evaluated the usage of new therapeutic drug mavacamten in the treatment of obstructive HCM, EXPLORER and PIONEER HCM, used the change in peak VO2 as either both primary or secondary endpoint, or only secondary endpoint, indicates the importance of functional testing in management of HCM. (26, 27).

Reviewer 2 Report

Summary

The article is a literature review on the pathophysiology of hypertrophic cardiomyopathy (HMC), the interpretation of various parameters of cardio-pulmonary exercise testing (CPET) in this condition, and the presentation and discussion of the prognostic value of data obtained in various research studies over the past decades. It reviews international recommendations on the use of cardiopulmonary exercise testing (CPET) in hypertrophic cardiomyopathy and its utility in the differential diagnosis of athlete’s hypertrophic cardiomyopathy , as well as in exercise prescription for these HMC patients.

General concept comments

The article is very interesting, and the review is comprehensive. It is true that its structure and content are similar to the article number 4 referenced by the authors in the bibliography ("Magrì D, Santolamazza C. Cardiopulmonary Exercise Test in Hypertrophic Cardiomyopathy. Ann Am Thorac Soc. 2017 Jul;14(Supplement_1):S102-S109. doi: 10.1513/AnnalsATS.201611-884FR. PMID: 28375659"), but being 6 years later in time, it allows for updates and includes 20 new bibliographic citations from studies conducted in recent years. Although there is still a lack of evidence regarding the prognostic value of CPET in hypertrophic cardiomyopathy, these types of publications and the new studies that arise from them may modify the (already changing) recommendations of clinical guidelines in the coming years.

Review

The article is organized into different sections that I consider appropriate because they facilitate reading. However, I suggest a reorganization of the content with the creation of a specific section on "recommendations/prescription of physical exercise in hypertrophic cardiomyopathy" that includes the lines written by the authors 294-303 and 328-333.

The exposition is adequate, and the conclusions are correct.

The bibliography is appropriate. Perhaps the authors did not have time to include this recent and relevant article by "Robyn Bryde, Matthew W. Martinez, Michael S. Emery, Exercise recommendations for patients with hypertrophic cardiomyopathy, Progress in Cardiovascular Diseases, 2023."

Specific comments:

Line 33: In the keywords, I suggest removing the ";" and changing "Cardiopulmonary; exercise testing; hypertrophic cardiomyopathy" to "Cardiopulmonary exercise testing; hypertrophic cardiomyopathy".

Line 68: I suggest changing "increase LV pressure" to "increase LV telediastolic pressure".

Line 101: Change "amount of oxygen pumped by the heart in each cycle" to "amount of oxygen extracted by the tissues per heartbeat".

Line 217: The unexplained acronym "NMRI" can be removed in this line since it is explained in the following line.

Line 260: Change "suggest exercise testing" to "suggest CPET imaging or exercise stress echocardiography".

Author Response

Reviewer 2

Summary

The article is a literature review on the pathophysiology of hypertrophic cardiomyopathy (HMC), the interpretation of various parameters of cardio-pulmonary exercise testing (CPET) in this condition, and the presentation and discussion of the prognostic value of data obtained in various research studies over the past decades. It reviews international recommendations on the use of cardiopulmonary exercise testing (CPET) in hypertrophic cardiomyopathy and its utility in the differential diagnosis of athlete’s hypertrophic cardiomyopathy, as well as in exercise prescription for these HMC patients.

General concept comments

The article is very interesting, and the review is comprehensive. It is true that its structure and content are similar to the article number 4 referenced by the authors in the bibliography ("Magrì D, Santolamazza C. Cardiopulmonary Exercise Test in Hypertrophic Cardiomyopathy. Ann Am Thorac Soc. 2017 Jul;14(Supplement_1):S102-S109. doi: 10.1513/AnnalsATS.201611-884FR. PMID: 28375659"), but being 6 years later in time, it allows for updates and includes 20 new bibliographic citations from studies conducted in recent years. Although there is still a lack of evidence regarding the prognostic value of CPET in hypertrophic cardiomyopathy, these types of publications and the new studies that arise from them may modify the (already changing) recommendations of clinical guidelines in the coming years.

Review

  1. The article is organized into different sections that I consider appropriate because they facilitate reading. However, I suggest a reorganization of the content with the creation of a specific section on "recommendations/prescription of physical exercise in hypertrophic cardiomyopathy" that includes the lines written by the authors 294-303 and 328-333.

We are very thankful for this insight. We agree that in this way the prescription of physical activity to patients with HCM is apostrophized and better recognized as an important issue.

Page 7

  1. Recommendations for exercise prescription in hypertrophic cardiomyopathy

Previously, avoidance of competitive sports in HCM patients was strongly recommended. However, there are some new perspectives that are suggesting that, in some cases, participation in competitive sports can be safely managed. More data is needed to provide an objective safety evaluation and decision making, for example after septal reduction procedures or ICD implantation have been done (17). The ESC Preventive Cardiology guidelines suggest individual decision making and recommend (Class IIc) exercise testing to assess for the appearance of exercise-induced ventricular tachycardia, a significant increase in LV outflow gradient (>50 mmHg) or ABPRE. These findings on an exercise test in conjunction with symptoms (i.e., syncope), high ESC risk score and history of SCD, represent absolute contraindications for competitive sports in athletes with HCM (45).

Whereas guidelines do not strictly point out the use of CPET for exercise prescription, evidence show that mild and moderate exercise programs in asymptomatic and symptomatic individuals with HCM are safe and lead to increased functional capacity and improved quality of life (48). Less data is available on the effects of vigorous exercise (48). This topic was discussed in the most recent article by Bryde and the authors, who emphasized the importance of physical activity in HCM management (49). In the study of Larsen et al, where almost half of the patients with HCM were overweight (45% of 510 patients had a BMI ≥ 30 kg/m2), the authors suggested that obesity can play a big attributing factor to reduced exercise capacity unrelated to the disease itself (50).  In this condition moderate physical activity at the same time helps weight loss, but also functional capacity improvement (50). Individuals with HCM who regularly exercise have lower total cardiovascular mortality compared with those who do not (48). 

The exposition is adequate, and the conclusions are correct.

  1. The bibliography is appropriate. Perhaps the authors did not have time to include this recent and relevant article by "Robyn Bryde, Matthew W. Martinez, Michael S. Emery, Exercise recommendations for patients with hypertrophic cardiomyopathy, Progress in Cardiovascular Diseases, 2023."

Thank you for drawing our attention to this great review article by Bryde and the authors. We included it in the section about physical exercise recommendations.

Line 363:

This topic was discussed in the most recent article by Bryde and the authors, who emphasized the importance of physical activity in HCM management (49).

  1. Specific comments:

Thank you for your recommendations. All suggested changes in the following lines have been made.

Line 32: In the keywords, I suggest removing the ";" and changing "Cardiopulmonary; exercise testing; hypertrophic cardiomyopathy" to "Cardiopulmonary exercise testing; hypertrophic cardiomyopathy".

Line 70: I suggest changing "increase LV pressure" to "increase LV telediastolic pressure".

Line 103: Change "amount of oxygen pumped by the heart in each cycle" to "amount of oxygen extracted by the tissues per heartbeat".

Line 233: The unexplained acronym "NMRI" can be removed in this line since it is explained in the following line.

This abnormal response was associated with higher functional disability, history of atrial fibrillation, NMRI signs of fibrosis at nuclear magnetic resonance imaging (NMRI) and the usage of antiarrhythmic drugs, especially beta-blockers (9).

Line 297: Change "suggest exercise testing" to "suggest CPET imaging or exercise stress echocardiography".

Reviewer 3 Report

Great abstract. I would recommend backing every statement in the abstract with an example from a registry or study with a large patient pool. For example, please perform a literature search to support that CPET with Echo is superior to CFT etc.

You may discuss this further by providing standard genetic profiles for each condition leaving to HCM. Please provide a number of patients when making statements throughout the paper. E.g., lines 66-67! approximate 2/3 of patients from what sample size? What age?  What comorbidities? This is not an accepted reference. There are several areas where there are statements, without proper details. Please fix them prior to final publications.

In some places you have mentioned that “studies show” with a single citation. Please elaborate on this “studies” or “study” with et. al format and more references and ranges. (e,g, 112-113)

Overall is clinically relevant that CPET may allow better imaging of HCM however this paper requires a significant revamp, and a more structured literature search for each statement made.

Please remove the template language at the end or update it as necessary. 

Author Response

Reviewer 3

  1. Great abstract. I would recommend backing every statement in the abstract with an example from a registry or study with a large patient pool.

For example, please perform a literature search to support that CPET with Echo is superior to CFT etc.

You may discuss this further by providing standard genetic profiles for each condition leaving to HCM.

Thank you for your meaningful suggestions in order to improve the quality of our review. We considered the abstract to be a relevant guide to the article itself where the references are made. We added following paragraphs and references:

  1. Future of CPET in HCM

Although performed in fewer institutions, CPET combined with echocardiography (CPET imaging) is more informative in terms of cardiac functional status, LV outflow tract obstruction and diastolic dysfunction during exercise in patients with HCM patients (15). It also allows post septal reduction follow up and monitoring the response to the procedure (15). There is a strong theoretical and practical rationale supporting the simultaneous gas exchange analysis with stress echocardiography. (51, 52, 53) They provide simultaneously a comprehensive understanding of overall functional status in the light of contractile function and relaxation of cardiac chambers, valve function and left-and right-sided cardiac hemodynamics, both enabling strong diagnostic and prognostic information. (54) Huge advantage of this method is a noninvasive cardiac output calculation, which can be critical for adequate clinical insight in patients with HCM. (54) The appearance of B-lines in the lungs, detected by echocardiography during CPET, is a marker of pulmonary congestion during effort, as shown in 2/3 of total 128 patients diagnosed with HCM examined by Palinkas ED et al. (55) In the study of Re F et al, exercise induced pulmonary hypertension was present in about one fifth of total 182 patients diagnosed with HCM without evidence of elevated pulmonary pressures at rest and was associated with adverse clinical outcomes, suggesting that echocardiography combined with CPET may help physicians to detect early stage of pulmonary hypertension thus allowing a closer clinical monitoring and individualized therapies. (56) Another proven advantage of combined stress echocardiography and CPET is the ability to assess the right heart - pulmonary circulation uncoupling which is likewise relevant in patients with pulmonary hypertension associated with HCM. (57) The estimation of right ventricular contractile reserve by echocardiography may play important role for the follow-up and therapeutic management of patients with HCM complicated with pulmonary hypertension (57). Yet, large clinical trials are to be conducted.

The role of genetic testing over the CPET variables in the HCM risk stratification was analyzed in the retrospective genotype-phenotype study of 371 patients screened for likely pathogenic/pathogenic (LP/P) genetic variants, at least for the main sarcomeric genes MYBPC3 (myosin binding protein C), MYH7 (β-myosin heavy chain), TNNI3 (cardiac troponin I) and TNNT2 (cardiac troponin T). (58) The LP/P variant was associated with a more aggressive HCM phenotype, however, left atrial diameter, circulatory power (peak VO2*peak systolic blood pressure) and ventilatory efficiency were the only independent predictors of HF, whereas only left atrial size and circulatory power were predictors of the sudden cardiac death after 5.1 years of follow up. This finding reaffirms pivotal role of the CPET-echo-derived parameters in the HCM risk stratification, and indicates that the combination of genetic testing and CPET-echo may have adjunctive role in prognosis of patients with HCM. (58)

 Invasive CPET with the measurement of pulmonary pressures allows a better assessment of exercise induced diastolic dysfunction in HCM (20). Simultaneous cardiac catheterization and CPET can provide essential hemodynamic data and identify patients who might benefit from septal reduction therapy but can also determine the presence of comorbidities, enable monitoring of the response to medication and surgical interventions, estimate prognosis, and guide referral for orthotopic heart transplantation. (59) Typical hemodynamic features predicting a favorable response to surgery include left atrial hypertension secondary to dynamic mitral regurgitation and acute development of LV outflow tract obstruction. (60) The invasive assessment of the hemodynamic adaptations to effort combined with CPET, in patients with HCM, may provide greater insight into the primary mechanism of exertional symptoms, which might better predict response to septal reduction therapy since it is a physiological rather than pharmacological provocation. (61) However, there are challenges performing the accepted, more invasive, catheter-enabled measures in many clinical settings due to logistical complications, patient risk and excessive cost. Equipping more laboratories around the world in this regard may contribute to better diagnosis and management of HCM.

  1. Borghi-Silva A, Labate V, ArenaR, et al.: Exercise ventilatory power in heart failure patients: functional phenotypes definition by combining cardiopulmonary exercise testing with stress echocardiography. Int J Cardiol. 2014; 176:1348–1349. 
  2. Bandera F, Generati G, Pellegrino M, et al.: Role of right ventricle and dynamic pulmonary hypertension on determining ΔVO2/ΔWork Rate flattening: insights from cardiopulmonary exercise test combined with exercise echocardiography. Circ Heart Fail. 2014;7(5):782-790.
  3. Re F, Zachara E, Avella A, et al.: Rest and latent obstruction in hypertrophic cardiomyopathy: impact on exercise tolerance. J Cardiovasc Med (Hagerstown). 2013;14(5):372-379.
  4. Guazzi M, Arena R, Halle M, et al.: 2016 focused update: clinical recommendations for cardiopulmonary exercise testing data assessment in specific patient populations.  Eur Heart J. 2018;39(14):1144-1161.
  5. Pálinkás ED, Re F, Peteiro J, et al. : Pulmonary congestion during Exercise stress Echocardiography in Hypertrophic Cardiomyopathy.  Int J Cardiovasc Imaging. 2022;38(12):2593-2604.
  6. Re F, Halasz G, Moroni F, et al.: Exercise-induced pulmonary hypertension in hypertrophic cardiomyopathy: a combined cardiopulmonary exercise test-echocardiographic study. Int J Cardiovasc Imaging. 2022;38(11):2345-2352.
  7. Guazzi M, Villani S, Generati G, et al.: Right Ventricular Contractile Reserve and Pulmonary Circulation Uncoupling During Exercise Challenge in Heart Failure: Pathophysiology and Clinical Phenotypes. JACC Heart Fail. 2016;4(8):625-635.
  8. Magrì D, Mastromarino V, Gallo G, et al.: Risk Stratification in Hypertrophic Cardiomyopathy. Insights from Genetic Analysis and Cardiopulmonary Exercise Testing. J Clin Med. 2020;9(6):1636.
  9. Lanier GM, Fallon JT, Naidu SS: Role of Advanced Testing: Invasive Hemodynamics, Endomyocardial Biopsy, and Cardiopulmonary Exercise Testing. Cardiol Clin. 2019;37(1):73-82.
  10. Prasad M, Geske JB, Sorajja P, et al.:Hemodynamic changes in systolic and diastolic function during isoproterenol challenge predicts symptomatic response to myectomy in hypertrophic cardiomyopathy with labile obstruction. Catheter Cardiovasc Interv. 2016; 88:962–970.
  11. Caravita S, Baratto C, Perego GB, et al.: Invasive Hemodynamics of Hypertrophic Cardiomyopathy: Exercise Versus Isoproterenol. Circ Heart Fail. 2020;13(6):e007000.

  1. Please provide a number of patients when making statements throughout the paper. E.g., lines 66-67! approximate 2/3 of patients from what sample size? What age?  What comorbidities? This is not an accepted reference. There are several areas where there are statements, without proper details. Please fix them prior to final publications.

Thank you for your suggestions, in the following corrections we tried to be more specific.

Page 2

Line 65 That number rises to almost half of the patients during exercise - out of 11,672 patients studied from 69 different research articles, LVOT gradient > 30 mmHg was present at baseline in 31.4% of cases, and increased to 49% during exercise (7). Shah and colleagues found that 54 out of 87 symptomatic patients with no previously documented LVOT have latent LV outflow tract obstruction (8).

Page 3

Line 117 Analyzing more than 50 publications in HCM cohorts and collectively 11,672 HCM patients (48 ± 14 years old, of which 65.9% were men and 34.1% women), Bayonas-Ruiz  et al. found that mean peak VO2 was 22.3 ± 3.8 mLO2●kg-1●min-1 and concluded that it is a disease with reduced exercise capacity of at least 20% from what is expected for in an apparently healthy individual (7). …Masri et al. studied 1005 HCM patients (50±14 years old, 64% male) and found that a peak VO2 <50 % of predicted was independently associated with overall mortality and SCD (22). Similarly, in a study of 156 patients with HCM (mean follow-up 27±11 months), Finocchiaro et al. concluded that a peak VO2 <20 mLO2●kg-1●min-1 or <80% of predicted was associated with worse prognosis (23). Coats et al. followed 1898 patients with HCM (average age 47±15 years, 67% male) during the time period between 1998 and 2010, who underwent CPET at the beginning of the follow up period. In this study aerobic capacity was shown to be a strong predictor for death and heart transplantation (primary end points). The patients who had peak oxygen consumption ≤15.3 mL/kg/min had a 14% and 31% chance of dying or having a heart transplant in 5 year and 10 year period respectively  (16). In a prognostic study of Sorajja et al, with a 4.0±3.2 years follow up, population of 182 minimally symptomatic patients with obstructive HCM (mean age 53 ± 15 years; 65% were men, 35% women), who had peak VO2 less than 60% of the predicted value had 41% chance of dying or having severe symptoms in a 4 year period (24). The fact that in the earlier mentioned study by Masri et al. only 8% of patients with HCM achieved >100% age-gender predicted peak VO2 (22) can be a valuable clinical measure in the differentiation between physiologic LV hypertrophy in athletes and HCM, especially in the so called “gray zone” (25). Recently, 58 athletes with HCM evaluated at Mayo Clinic were found to have mildly reduced exercise capacity (83% of predicted), however, reduced peak VO2 did not correlate with symptom status or clinical outcomes (17).

Page 4

Line 168 In the study of Mapelli et al. 96 out of 312 patients with non-end-stage HCM (70% of which were males, age 49±18 years), thus 31%, presented with abnormal O2-pulse temporal behavior, irrespective of LVOTO values. 

Line 188 In a study cohort of 623 HCM patients (49±16 years old, 69% male, 3.7 years follow up), Magri et al investigated CPET parameters that can predict SCD risk, finding a VE/VCO2 slope >31 to be the most accurate among CPET parameters in predicting the SCD end point (sensitivity 64%, specificity 72%) (29). In earlier mentioned study by Coats at el, the risk of death or heart transplantation was increased by 18%...

Line 196 In a study by Velicki et al. who analyzed a total of 41 patients with nonobstructive HCM who were recruited into the ongoing SILICOFCM study the VE/VCO2 slope was found to be the most sensitive CPET parameter for gauging therapeutic efficacy of 16 weeks sacubitril/valsartan threatment (30).

Page 5

Line 231 Efthimiadis et al. studied 68 HCM patients (age 44.8± 14.6 years, 45 males), who underwent CPET and discovered that half of the cohort demonstrated chronotropic incompetence (9). … During 4.2 years follow up of 681 patients with HCM (48 ± 16 years old, 68% male), Magri et al found that a peak HR < 70% was a risk factor for HF related events and found this abnormal response held prognostic significance in HCM patients (32).

Line 247 The occurrence of arrhythmia is higher in young adults with HCM and the presence of ventricular arrhythmias is significantly associated with adverse events, showed the data metanalysis by Bayonas-Ruiz et al(7).

Line 250  Gimeno et al. studied 1380 patients with HCM (42 ± 15years; 62% male; mean follow-up 54 ± 49 months). Twenty-seven patients (mean age 40± 14 years) had NSVT (24) or ventricular fibrillation (VF) (3) during exercise. These patients had more severe hypertrophy and larger left atria. Occurrence of NSVT/VF was more common in males (22 (81.5%) were male)

Page 6

Line 270 Potratz et al. studied 132 patients with hypertrophic non-obstructive cardiomyopathy (HNCM) (51 ± 17 years old; male 71%) who…

Line 281 As previously mentioned, Bayonas-Ruiz et al. analyzed 69 publications including 11,672 patients with HCM and they showed that LVOT was present at rest in 31.4% of cases, and increased to 49% during exercise (7). A LV outflow tract gradient of ≥50 mmHg is hemodynamically important, usually causes symptoms and thus, represents a threshold in treatment decision making regarding invasive septal reduction procedures (39). Sorraja et al. recognized the severity of LV outflow tract gradient at rest as an individual prognostic factor for death and severe adverse events after 4.0±3.2 years follow up of 182 patients with minimaly symptomatic HOCM (24). In this study LVOT obstruction at rest was present in 96 patients (53%), and latent in 86 patients (24). Interestingly, in the study of Mapelli et al. patients were grouped by the degree of LVOT (72% of patients with LVOTO <30; 10% between 30 and 49 and 18% ≥50 mm Hg) and the results showed that neither of the three analyzed CPET parameters (% of predicted peak VO2, O2-pulse and VE/VCO2 slope) was correlated to the degree of LVOTO.

  1. In some places you have mentioned that “studies show” with a single citation. Please elaborate on this “studies” or “study” with et. al format and more references and ranges. (e,g, 112-113)

The reffered line (112-113) was adjusted, but incorporated in other section the the review – line 364.

In the study of Larsen et al, where almost half of the patients with HCM were overweight (45% of 510 patients had a BMI ≥ 30 kg/m2), the authors suggested that obesity can play a big attributing factor to reduced exercise capacity unrelated to the disease itself (50).

  1. Overall is clinically relevant that CPET may allow better imaging of HCM however this paper requires a significant revamp, and a more structured literature search for each statement made. Please remove the template language at the end or update it as necessary. 

According to valuable Reviewers suggestions we have made an extensive edit, supported with more literature, as shown above. The text at the end is changed. We sincerely hope that our work is now suitable for publication.
